# Type 2 Diabetes Mellitus and COVID-19: A Narrative Review

**DOI:** 10.3390/biomedicines10092089

**Published:** 2022-08-26

**Authors:** Cristina Rey-Reñones, Sara Martinez-Torres, Francisco M. Martín-Luján, Carles Pericas, Ana Redondo, Carles Vilaplana-Carnerero, Angela Dominguez, María Grau

**Affiliations:** 1Research Support Unit-Camp de Tarragona, Catalan Institute of Health (ICS), 43005 Tarragona, Spain; 2IDIAP Jordi Gol, Catalan Institute of Health (ICS), USR Camp de Tarragona, 43202 Reus, Spain; 3School of Medicine and Health Sciences, Universitat Rovira i Virgili, 43201 Reus, Spain; 4Department of Medicine, University of Barcelona, 08036 Barcelona, Spain; 5Hospital Universitario Bellvitge, Catalan Institute of Health (ICS), 08907 Barcelona, Spain; 6IDIAP Jordi Gol, Catalan Institute of Health (ICS), 08007 Barcelona, Spain; 7Biomedical Research Consortium in Epidemiology and Public Health (CIBERESP), 28029 Madrid, Spain; 8Serra Húnter Fellow, Department of Medicine, University of Barcelona, 08036 Barcelona, Spain; 9August Pi i Sunyer Biomedical Research Institute (IDIBAPS), 08036 Barcelona, Spain

**Keywords:** type-2 diabetes mellitus, epidemiology, COVID-19, bidirectional link, antidiabetic treatment

## Abstract

Type-2 diabetes mellitus (T2DM) is a chronic metabolic disorder. The incidence and prevalence of patients with T2DM are increasing worldwide, even reaching epidemic values in most high- and middle-income countries. T2DM could be a risk factor of developing complications in other diseases. Indeed, some studies suggest a bidirectional interaction between T2DM and COVID-19. A growing body of evidence shows that COVID-19 prognosis in individuals with T2DM is worse compared with those without. Moreover, various studies have reported the emergence of newly diagnosed patients with T2DM after SARS-CoV-2 infection. The most common treatments for T2DM may influence SARS-CoV-2 and their implication in infection is briefly discussed in this review. A better understanding of the link between TD2M and COVID-19 could proactively identify risk factors and, as a result, develop strategies to improve the prognosis for these patients.

## 1. Introduction

In the last decades, type-2 diabetes mellitus (T2DM) has become a chronic metabolic disorder caused by the interaction of different genetic and environmental factors. The incidence and prevalence of patients with T2DM are increasing worldwide, even reaching epidemic values in most high- and middle-income countries [1]. The World Health Organization (WHO) estimates that T2DM will be the seventh leading cause of death by 2030 worldwide (Figure 1) [2]. The main reasons for this increase seem the high prevalence of obesity and the unhealthy lifestyles. Uncontrolled and prolonged T2DM can lead to serious complications, some of them being life-threatening [3]. As a result, the healthcare cost of T2DM and the related diseases is growing every year [4]. Strategies to control T2DM include appropriate life-style changes as well as medication intake when necessary [5].

T2DM could be a risk factor for developing complications in other diseases. At the onset of the pandemics, the USA Centers for Disease Control and Prevention (CDC) described that one third of patients infected with COVID-19 had comorbidities. Thus, people with at least one underlying condition account for 78% of admissions to the intensive care unit (ICU) and 94% of deaths. T2DM was the most frequently reported, being the 10.9% of the cases [6]. In addition, a fast-growing evidence reports a bidirectional interplay between T2DM and COVID-19. Clinical data so far suggest that the severe acute respiratory syndrome coronavirus 2 (SARS-CoV-2) infection may result in metabolic dysregulation and in impaired glucose homeostasis [7].

The objective of this review is to provide an overview of the most recent studies that point to T2DM as a risk factor and poor prognosis of COVID-19, as well as, to summarize the potential mechanisms involved in this relationship.

## 2. T2DM as a Risk Factor for the Development and Prognosis of COVID-19 

Several risk factors have been associated with an increased risk of SARS-CoV-2 infection and complication. For instance, male sex, older age, deprivation and comorbidities such as cardiopathy, hypertension, chronic obstructive pulmonary disease, immunosuppression or T2DM [8,9]. Thus, 33.8% of 5700 of patients with COVID-19 admitted to 12 hospitals within the Northwell Health system in New York had T2DM [10]. In addition, a random meta-analysis of 18 different studies determined that the risk of severe disease was 2.4-fold higher in patients with T2DM compared with those without [11], whereas another one showed a 2.6-fold higher severity risk by increased fasting blood glucose at admission [12].

### 2.1. Risk of Death and Complications in T2DM Patients with COVID-19 

So far, different studies have shown that people with T2DM have higher risk of COVID-19 mortality compared with non-diabetic individuals [11,12,13,14,15,16,17,18,19]. Nevertheless, Al-Salameh et al., reported that COVID-19-related death in patients with T2DM was lower than in general population, but the rate of intensive care unit (ICU) admission was increased [20]. As highlighted by Diedisheim et al., the age may play a key role because after age 50 years, diabetes-related risk might be weakened by all other comorbidities or conditions associated with aging [15] (Table 1).

In addition to the increased mortality associated with COVID-19, patients with T2DM also present with more complications from such infection, even requiring admission to the ICU or dying [10,20]. Moreover, uncontrolled hyperglycemia can also be a risk factor for an adverse COVID-19 prognosis. An observational study including more than 1000 patients hospitalized with COVID-19 in USA showed that 40% of patients had diabetes or uncontrolled hyperglycemia at admission, and hospital mortality was four times higher for DM patients. The same study showed that mortality was seven times higher for those without pre-existing T2DM who developed in-hospital hyperglycemia [14]. Similarly, a recent meta-analysis that included 14,502 patients confirmed these findings and showed a nonlinear relationship between fasting blood glucose at admission and severity: every mmol/L enhancement in glucose levels increased the risk of COVID-19 severity by 33% [12].

### 2.2. Potential Mechanisms Underlying Unfaborable Clinical Outcomes of COVID-19 in People with Diabetes 

Epidemiological studies have determined the severity of COVID-19 due to a number of complications or comorbidities associated with T2DM. Co-occurrence of microvascular and macrovascular T2DM complications, including cardiovascular disease, renal failure, retinopathy and reduced renal function, could be responsible for the increased poor COVID-19 outcomes and mortality after infection [21,22,23].

T2DM, even in the stages of prediabetes, is characterized by a dysregulation of glucose homeostasis, chronic inflammatory and prothrombotic state accompanied with other affectations, such as metabolic, vascular, immune and hematological abnormalities, which could explain the negative response to infections [24]. Some of these alterations have been proposed to explain T2DM impact on COVID-19 prognosis, including glucotoxicity, endothelial damage, chronic inflammatory state, oxidative stress and abnormal cytokine production [25,26].

Hyperglycemia could directly exacerbate SARS-CoV-2 infection, promoting the expression and activation of, angiotensin-converting enzyme 2 (ACE2) cellular receptor, the main receptor of the SARS-CoV-2, and increasing the expression of the serine protease TMPRSS2, which mediates the cleavage of the viral spike protein [27]. Of note, high glucose levels increase the production of inflammatory cytokines and cellular mediators and pro-thrombotic processes, promoting the development of acute cardiovascular complications [28]. Moreover, chronic hyperglycemia could compromise the innate and humoral immune response inhibiting lymphocyte proliferation, reducing the activity of natural killer cells and affecting the function of monocyte/macrophage and neutrophils [24,29]. According to this, different reports demonstrated that elevated glucose levels in admission is an independent risk factor for critical progression and in-hospital mortality in COVID-19 patients [30,31,32,33]. Therefore, uncontrolled hyperglycemia takes part to other COVID-19 complications, such as atherosclerosis, diabetic nephropathy, peripheral arteriosclerosis, and diabetic ketoacidosis [30]. Thus, hyperglycemia management has been proposed to improve clinical COVID-19 outcomes.

Other evidences suggested that chronic endothelial dysfunction predisposes to severe COVID-19 disease. In this regard, hyperglycemia and insulin resistance leads to endothelial dysfunction and glycocalyx damage in patients with T2DM, leading to leucocyte adhesion and promoting procoagulant and antifibrinolytic state [34,35]. A recent study of in-hospital COVID-19 patients from China reported that the COVID-19 severity was correlated with increased blood levels of IL-6 and lactate dehydrogenase (LDH) [36]. Of note, patients with T2DM have a higher inflammatory response, mainly characterized by increased levels of interleukin-6 (IL-6), interleukin-2 (IL-2) and the tumor necrosis factor α (TNF-α) [37]. This fact could explain the rapid COVID-19 progression and severity in patients with T2DM.

Altogether, these alterations could explain why patients with T2DM have a worse prognosis of COVID-19 (Figure 2).

## 3. SARS-CoV-2 Infection as a Risk Factor of Morbidity and Mortality: Metabolic Deregulation and Homeostasis Alteration

Infection affects several pathophysiological pathways, during the course of the disease, which eventually leads to late complications. So far, clinical data suggest that SARS-CoV-2 may cause metabolic dysregulation and impairment of glucose homeostasis. A study suggests that infection may be a precipitating factor for acute hyperglycemia, worsening prognosis in poorly controlled T2DM patients [38]. Moreover, some studies focus on the role of glycemic control as a critical factor to reduce complications, severe outcomes and mortality during SARS-CoV-2 infection [39,40].

Other studies identified the endothelium, which expressed both the ACE2 receptor and the serine protease TMPRSS2, as a first key player on the homeostasis alteration. In healthy individuals, endothelium is considered to be major contributor to various physiological processes supporting homeostasis. Lambadiari et al. showed that SARS-CoV-2 can cause endothelial and vascular dysfunction, which was associated with impaired cardiac performance for four months after SARS-CoV-2 infection [41]. After infection, there is an increase of cytokine levels and immune cells, which could induce insulin resistance and hyperglycemia [42]. Similarly, studies about Severe Acute Respiratory Syndrome (SARS) and Middle Eastern Respiratory Syndrome (MERS) point out how inflammatory cells may affect the liver, altering insulin-mediated glucose uptake, resulting in hyperinsulinemia and hyperglycemia [43].

Other studies suggest that there are possible underlying mechanisms that could explain the acute damage of pancreatic islets by SARS-CoV-2 and the consequent loss of insulin secretory capacity [40,44]. The exacerbated immune response through the virus-mediated release of chemokines and cytokines could also damage pancreatic cells and impair their ability to detect glucose and release insulin. Moreover, the immune response of the virus can further affect the ability of the liver and muscles to identify alterations [40,44]. Considering previous experience with inflammatory responses, COVID-19 inflammatory and viral immune responses can affect insulin sensitivity and deregulate glucose metabolism, leading to a vicious cycle of hyperglycemia and inflammatory response that destroys tissue integrity and physiological function during critical stages of infection. Thus, frequently used drugs in the treatment of COVID-19, such as corticosteroids or antiviral agents, can aggravate hyperglycemia and result in lipodystrophy and insulin resistance [45,46].

Regarding glucose homeostasis, severe SARS-CoV-2 infection contributes to insulin resistance and hyperglycemia by increased cytokines and unregulated compensatory hormonal response [21,47]. Moreover, glucotoxicity and the associated consequences, added to the increase of inflammatory cytokines by SARS-CoV-2 infection, oxidative stress, immune dysfunction and endothelial damage, predict an increase in metabolic complications, an increased risk of thromboembolism and multiorgan damage in individuals with T2DM [25,48].

## 4. Newly Emerging Patients with T2DM Infected with SARS-CoV-2

SARS-CoV-2 infection would have a direct effect of on glucose metabolism. A recent metanalysis have shown 492 cases of newly diagnosed diabetes from eight studies that included 3711 COVID-19 patients with a pooled proportion of 14.4% (95% confidence interval [5.9–25.8%]) [49]. Recent findings suggest a direct effect of SARS-CoV-2 on glucose metabolism resulting in new presentations of T2DM characterized by diabetic ketoacidosis, hyperosmolarity and unusually high insulin requirements to achieve glycemic control [49,50]. Although the evidence of a direct relationship between how SARS-CoV-2 is still sparse, Kazakou et al. described to possible links. First, the SARS-CoV-2 infection could damage B cells in the pancreas through the direct cytolytic effect of the virus. Second, the direct damage to the endocrine system during infection, which could contribute to the development of glucose and metabolic abnormalities in people previous infected (Figure 3) [42]. Understanding the effects of COVID-19 on glucose metabolism and homeostasis is essential to prevent and control complications associated with this infection and to help patients recovery [7].

## 5. How Antidiabetic Treatment Influences SARS-CoV-2 Infection

Blood glucose control may be crucial as a preventive measure for adverse outcomes related to COVID-19 [7]. Table 2 shows the influence of the most common treatments for T2DM on SARS-CoV-2 infection.

### 5.1. Metformin

As discussed above, inflammatory exacerbation due to increased cytokines levels has been recognized as one of the main keys to poor prognosis in COVID-19. The anti-inflammatory properties of metformin are already known, suggesting a beneficial effect for COVID-19 disease [52]. Despite this, immunomodulatory actions of metformin in the context of COVID-19 remain unclear. Some evidence point out the positive implications of metformin in COVID-19 as reduced insulin resistance and inhibition of virus entry through AMPK activation and phosphorylation of ACE2 [7]. Crouse et al., affirmed that although T2DM is a factor risk for COVID-19-related mortality, the risk is drastically reduced in subjects treated with metformin before the diagnosis of COVID-19, considering that metformin can provide a protective effect in this high-risk population [53]. In addition, a meta-analysis estimated a risk reduction associated with this treatment in individuals with T2DM and SARS-CoV-2 infection (Odds ratio (OR) = −0.37; 95% CI [−0.59; −0.16]) [54].

According to the practical recommendations for the management of T2DM in patients with COVID-19 a treatment cessation with metformin is recommended in dehydrated patients and follow the guidelines according to the infection, because lactic acidosis is likely to appear [55]. The same guide recommends careful surveillance of kidney function because of the high risk of chronic kidney disease or acute kidney damage [27].

### 5.2. Insulin

Insulin treatment is the first treatment option for controlling hyperglycemia in critical patients. Available evidence suggests that it is also the most appropriate hypoglycemic agent in hospitalized patients with severe T2DM and COVID-19 [56]. Recent studies have shown that severe cases of newly onset or pre-existing T2DM patients with COVID-19, high doses of intravenous insulin infusion are needed to control glycemic levels [42]. According to this, continuous glucose monitoring during hospitalization helps to ensure good control and minimize the risk of hypoglycemia in people on insulin therapy. Thus, insulin treatment results in an improvement in glycemic control in patients hospitalized in the ICU and offers the advantage of remote monitoring [57,58].

Some evidences indicate that patients with T2DM under insulin treatment for COVID-19 present worse clinical outcomes when compared with other antidiabetic drugs in early-stage of the disease. In fact, insulin treatment was associated with increased systemic inflammation and increased damage to vital organs, suggesting that insulin therapy for patients with COVID-19 and T2DM needs to be used with caution (27.2% versus 3.5%; adjusted Hazard Ratio = 5.38; 95% CI [2.75–10.54]) [59]. Thus, insulin treatment reflects a more advanced stage of the disease with associated co-morbidities, indicating the need of studies in this regard [60]. However, insulin remains the star treatment in serious hospitalized patients [27,48].

### 5.3. SGLT2 (Sodium-Glucose Cotransporter-2) Inhibitors 

SGLT2 inhibitors are a type of oral medication indicated to reduce blood glucose in adults with T2DM. Patients treated with SGLT2 inhibitors have an increased risk of dehydration and euglycemic diabetic ketoacidosis in case of acute illness, especially if the disease accompanies anorexia and vomiting [61]. Some studies suggest that SGLT2 inhibitors may reduce viral load due to increased lactate concentrations and decreased intracellular pH. In addition to the already known anti-inflammatory properties, primarily on endothelial function, SGLT2 inhibitors therapy could play a protective role in COVID-19-related organ failure [25]. The meta-analysis performed by Nguyen et al. showed a significant mortality reduction in individuals with T2DM treated with SGLT2 inhibitors previous to hospital admission (OR = 0.60 [0.40–0.88]) [62]. Nevertheless, a double-blind, placebo-controlled clinical trial with dapagliflozin in COVID-19 hospitalized patients with one or more cardiometabolic risk factors did not show a significant reduction in organ dysfunction or death, or in clinical recovery [63]. The recommendations for the practice in the management of T2DM in patients with COVID-19 suggest the discontinuation of treatment with SGLT2 inhibitors in severely affected patients with COVID-19 and at risk of dehydration [27].

### 5.4. Sulfonylureas

Sulfonylureas are the oldest oral type of antidiabetic drugs. Sulfonylureas bind to the ATP-sensitive potassium channels on the pancreatic β-cells, resulting in membrane depolarization and, therefore, stimulating insulin secretion [64]. On the one side, a systematic review and meta-analysis highlight the potential for sulfonylurea treatment to reduce the risk of mortality in T2DM patients with COVID-19 (OR = 0.80; 95% CI [0.66; 0.96]) [54]. On the other side, as Drucker’s review points out, patients with T2DM and severe COVID-19 disease should avoid treatment with sulfonylureas and discontinue treatment in case of hospitalization due to the risk of hypoglycemia, especially in situations of low oral food intake or in combination with chloroquine or hydroxychloroquine treatments [48].

### 5.5. Thiazolidinediones

These drugs act as partial or selective agonists of the peroxisome activated by proliferator-γ (PPAR-γ) receptor, a nuclear receptor that regulates the transcription of several genes involved in glucose and lipid metabolism. Thiazolidinediones have shown to improve insulin resistance and have anti-inflammatory properties and anti-atherosclerotic effects, having the potential to mediate the protective effects of the cardiovascular system [25]. This treatment has been associated with weight gain and swelling, which has also led to an increase in heart failure. Therefore, its use is not recommended in patients with COVID-19, although the literature suggests that more clinical trials are needed to maximize the risk-benefit relationship of thiazolidinediones use in patients with COVID-19 [65].

### 5.6. Dipeptidyl Peptidase-4 (DPP4) Inhibitors

DPP4 is a transmembrane glycoprotein expressed in the spleen, lung, liver, kidneys, and immune cells or soluble circulating in the bloodstream. DPP4 has a crucial role in glucose homeostasis. DPP4 inhibitors in patients with DM2 have been used for a long time since 2006, with good tolerance and few adverse reactions reported [66]. The role of DPP-4 inhibition as an inflammation mitigator and potent antifibrotic agent is supported by several experimental studies [67].

Recent studies suggest that DPP4 inhibitors could benefit the treatment of mild or even severe cases of COVID-19. A metanalysis including 10 observational studies found that DDP-4 inhibitors reduce the risk for COVID-19-related mortality by 50% [67]. Other metanalysis also support the hypothesis that DDP-4 inhibitors could have a protective effect on COVID-19 (OR = 0.58; 95% CI [0.34–0.99]) [68]. Finally, the reduction in mortality was marginally significant in the metanalysis performed by Kan et al. (OR = 0.72; 95% CI [0.51–1.01]) [54]. Thus, further research is necessary to evaluate the role of DPP4 inhibitors in patients with T2DM and COVID-19 [69].

### 5.7. Glucagon-Like Peptide 1 (GLP-1) Receptor Agonist

GLP-1 is an incretine hormone responsible of blood glucose reduction through its receptor that reduce blood. Beyond controlling glycemia, GLP-1 receptor agonist has great potential for treating hyperglycemia [70]. Indeed, a review concluded that GLP-1 receptor agonist was able to reduce blood glucose levels and in turn, the administration of insulin without increasing the incidence of hypoglycemia. A recent Bayesian Network Metanalysis GLP-1 receptor agonist treatment was liked to a decrease in COVID-19-related mortality in T2DM individuals compared to non-users (OR = 0.91; 95% CI [0.84; 0.98]) [71]. However, there is not sufficient evidence to recommend GLP-1 receptor agonist for critical patients with T2DM and COVID-19 [72]. Regarding the adverse effects, this treatment can trigger gastrointestinal side effects such as nausea and vomiting, and consequently, aspiration in this type of patients. For this reason, GLP-1 receptor agonists are not recommended for patients with mild or moderate COVID-19 [27]. Among the most outstanding findings about GLP-1 receptor agonists is the reduction of major cardiac events in patients with T2DM, and its anti-inflammatory action under conditions of low-grade inflammation such as atherosclerosis and non-alcoholic fatty liver disease [73,74].

## 6. Summary

This review provides the latest insights into the interaction between COVID-19 and T2DM, which is considered a major risk factor for COVID-19 disease. Furthermore, new data have identified hyperglycemia and insulin resistance in patients after SARS-CoV-2 infection, leading to an emerging form of T2DM. The pathophysiological effects of both diseases (e.g. inflammation, immune response, endothelial damage and glucotoxicity) have been proposed as the main potential mechanisms behind this bidirectional interplay. Moreover, research works have identified the interaction between several T2DM treatments and the prognosis of COVID-19, leading to more specific recommendations for the treatment of hyperglycemia during COVID-19. A better understanding of the link between TD2M and COVID-19 could proactively identify risk factors and, as a result, develop strategies to improve the prognosis for these patients.

## Figures and Tables

**Figure 1 biomedicines-10-02089-f001:**
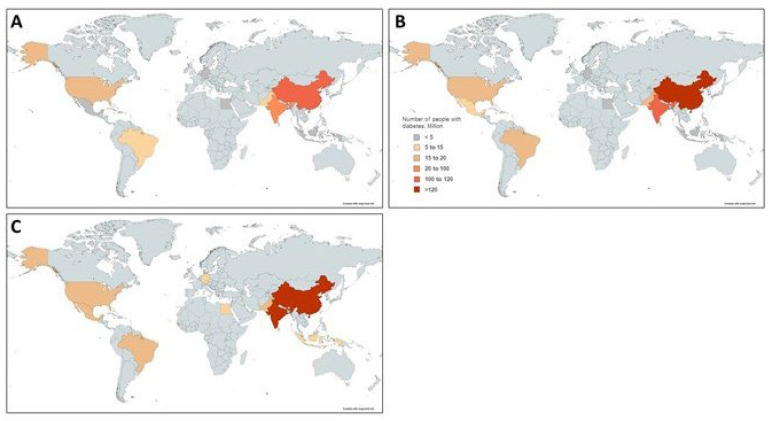
Estimated number of adults with diabetes (20–79 years) in the top 10 countries worldwide in (**A**) 2019, (**B**) 2030 and (**C**) 2045. Data for this figure is obtained from Mathers et al. [2].

**Figure 2 biomedicines-10-02089-f002:**
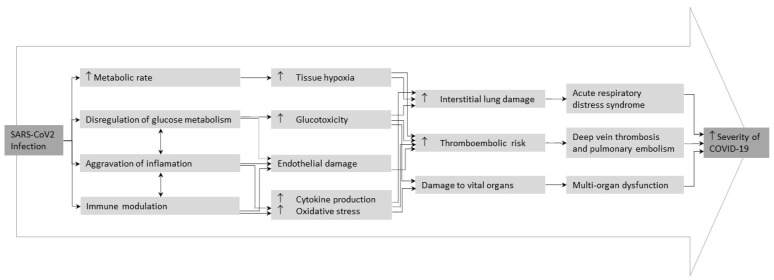
Possible mechanisms behind adverse clinical outcomes of COVID-19 in people with type 2 diabetes mellitus. Adapted from Lim et al. [25].

**Figure 3 biomedicines-10-02089-f003:**
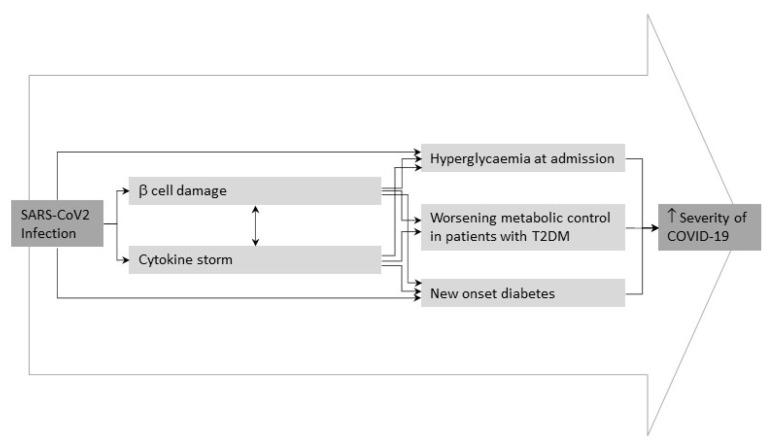
Potential pathogenic mechanisms of SARS-CoV-2 infection underlying metabolic deregulation and homeostasis alteration. Adapted from Apicella et al. [51].

**Table 1 biomedicines-10-02089-t001:** COVID-19 Death in T2DM: Summary of the main results.

	Country	*n*	Mortality (%)		COVID-19 Death in T2DM
			No T2DM	T2DM	HR (95% CI)
Al-Salameh [20]	France	433	21.5	17.4	0.77 (0.44–1.32)
Barron [13]	UK	61,414,470	0.03	0.26	2.03 (1.97–2.09)
Bode [14]	USA	1122	6.2	28.8	--
De Almeida-Pititto [11]	Meta-analysis	4,305	12.4	29.9	2.50 (1.74–3.59)
Diedisheim [15]	France	6314	22	26	1.81 (1.14–2.87)
Espiritu [16]	Philippines	10,881	12.9	26.4	1.46 (1.28–1.68)
Kim [17]	USA	10,861	--	--	1.20 (1.08–1.32)
Lazarus [12]	Meta-analysis	14,502	--	--	1.81 (1.41–2.33)
Williamson [18]	UK	17,278,392	0.06	0.26	1.95 (1.83–2.08)
Wu [19]	China	44,672	2.3	7.3	--

CI, confidence interval. HR, Hazard Ratio. T2DM, type 2 diabetes mellitus.

**Table 2 biomedicines-10-02089-t002:** Beneficial and adverse effects and clinical recommendations of antidiabetic treatment for COVID-19.

Anti-Diabetic Treatment	Beneficial Effect	Adverse Effect	Recommendations
Metformin	Anti-inflammatory effect Reduction insulin resistance Inhibition virus entry	Lactic acidosis (kidney damage)	Avoid in dehydrated patients
Insulin	Continuous glycemic control		First treatment
SGLT2 inhibitors	Anti-inflammatory effects		Avoid in severely affected patients with COVID-19 and at risk of dehydration
Sulfonylureas	Anti-inflammatory effects	Risk of hypoglycemia	Avoid in severe COVID-19 disease and combination with chloroquine or hydroxychloroquine treatments
Thiazolidinediones	Insulin resistance improvement Anti-inflammatory and anti-atherosclerotic effects and effects	Weight gain and swelling Heart failure	Not recommended in patients with COVID-19
DPP4 inhibitors	Anti-inflammatory and antifibrotic effects		Mild and severe cases of COVID-19
GLP-1 receptor agonists	Anti-inflammatory effect Reduction cardiac events Control glucose homeostasis	Gastrointestinal side effects	Control of blood glucose levels in ICU hospitalized patients

## Data Availability

Data sharing not applicable.

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
