# Peer review of "Type 2 Diabetes Mellitus and COVID-19: A Narrative Review"

_biomedicines, 2022, doi:10.3390/biomedicines10092089_

Round 1

Reviewer 1 Report

This work reports a comprehensive review of the interplay between diabetes and, the course of illness and outcome of Covid-19. The authors considered works dealing with an analysis of statistics for cases with coexisting diagnoses, metabolic risk factors, and interaction with antidiabetic treatment drugs. This review can serve as an important source of the present-state information for future studies and I recommend its accepting after improving minor issues:

·       *  some comments on the interesting contradiction between sources highlighted in subsection 3.1 that maybe deserves more details as in the present state of manuscript: Ref. [18] deals with a smaller sample than Refs. [15] and [17], and this sample (both groups, with and without diabetes) is well-localized in age (median 72-73 years with a relatively small scattering of ages). The results reported in [18] do not contradict results of [15], where is highlighted that diabetes is increased the Covid-19 mortality as a valuable factor primarily for younger people and “after age 50 years, diabetes-related risk is weakened by all other comorbidities or conditions associated with age”. Thus, I suppose that the diabetes-age-related issues could be specially additionally discussed and, the sentences describing “contradictions” should be corrected to a more accurate form.

·     *    a misprint: “S” is missed in the header of section 5: should by SARS instead of SAR

Author Response

Reviewer 1
This work reports a comprehensive review of the interplay between diabetes and, the course of illness and outcome of Covid-19. The authors considered works dealing with an analysis of statistics for cases with coexisting diagnoses, metabolic risk factors, and interaction with antidiabetic treatment drugs. This review can serve as an important source of the present-state information for future studies and I recommend its accepting after improving minor issues:

·       *  some comments on the interesting contradiction between sources highlighted in subsection 3.1 that maybe deserves more details as in the present state of manuscript: Ref. [18] deals with a smaller sample than Refs. [15] and [17], and this sample (both groups, with and without diabetes) is well-localized in age (median 72-73 years with a relatively small scattering of ages). The results reported in [18] do not contradict results of [15], where is highlighted that diabetes is increased the Covid-19 mortality as a valuable factor primarily for younger people and “after age 50 years, diabetes-related risk is weakened by all other comorbidities or conditions associated with age”. Thus, I suppose that the diabetes-age-related issues could be specially additionally discussed and, the sentences describing “contradictions” should be corrected to a more accurate form.
Reply: We have rewritten the paragraph regarding the COVID-19 mortality risk in type 2 diabetes mellitus patients (#2 in the new version of the manuscript) to include a discussion about the role of age (lines 57-86):

“2. T2DM as a risk factor for the development and prognosis of COVID-19

Several risk factors have been associated with an increased risk of SARS-CoV2 in-fection and complication. For instance, male sex, older age, deprivation and comorbid-ities such as cardiopathy, hypertension, chronic obstructive pulmonary disease, im-munosuppression or T2DM (Baena-Díez J.M., J. Public Health (Oxf.) 2020; Baena-Díez J.M. Healthc. (Basel, Switzerland) 2021). Thus, 33.8% of 5,700 of patients with COVID-19 ad-mitted to 12 hospitals within the Northwell Health system in New York had diabetes (Richardson, S. JAMA. 2020). In addition, a random meta-analysis of 18 different studies determined that the risk of severe disease was 2.4-fold higher in patients with T2DM compared with those without (De Almeida-Pititto, B. Diabetol. Metab. Syndr. 2020); whereas another one showed a 2.6-fold higher severity risk by increased fasting blood glucose at admission (Lazarus, G. Diabetes Res. Clin. Pract. 2021).

2.1. Risk of death and complications in T2DM patients with COVID-19 

So far, different studies have shown that people with T2DM have higher risk of COVID-19 mortality compared with non-diabetic individuals (De Almeida-Pititto, B. Diabetol. Metab. Syndr. 2020; Lazarus, G. Diabetes Res. Clin. Pract. 2021; Barron, E. Lancet. Diabetes Endocrinol. 2020; Bode, B. J. Diabetes Sci. Technol. 2020; Diedisheim, M. J Clin Endocrinol Metab. 2021; Espiritu, A.I. Sci. Rep. 2021; Kim, T.S. Obesity (Silver Spring). 2021; Williamson, E.J. Nature 2020; Wu, Z. JAMA 2020). Nevertheless, Al-Salameh et al., reported that COVID-19-related death in patients with T2DM was lower than general population, but the rate of intensive care unit (ICU) admission was increased (Al-Salameh, A. Diabetes. Metab. Res. Rev. 2021). As highlighted by Diedisheim et al., the age may play a key role because after age 50 years, diabetes-related risk might be weakened by all other comorbidities or conditions associated with aging (Diedisheim, M. J Clin Endocrinol Metab. 2021) (Table 1).

In addition to the increased mortality associated with COVID-19, patients with T2DM also present with more complications from such infection, even requiring ad-mission to the intensive care unit (ICU) or dying (Richardson, S. JAMA. 2020; Barron, E. Lancet. Diabetes Endocrinol. 2020). Moreover, uncontrolled hyperglycemia can also be a risk factor for an adverse COVID-19 prognosis. An observational study including more than 1,000 patients hospitalized with COVID-19 in US showed that 40% of patients had diabetes or uncontrolled hyperglycemia at admission, and hospital mortality was four times higher for DM patients. The same study showed that mortality was seven times higher for those without pre-existing T2DM who de-veloped in-hospital hyperglycemia (Bode, B. J. Diabetes Sci. Technol. 2020). Similarly, a recent meta-analysis that included 14,502 patients confirmed these findings and showed a nonlinear relationship between fasting blood glucose at admission and severity: every mmol/L enhancement in glucose levels increased the risk of COVID-19 severity by 33% (Lazarus, G. Diabetes Res. Clin. Pract. 2021)”.

In addition, the new version of the manuscript includes a Table that summarizes the main results of the referenced studies.

Country

N

Mortality (%)

COVID-19 Death in T2DM

No T2DM

T2DM

HR (95% CI)

Al-Salameh

France

433

21.5

17.4

0.77 (0.44–1.32)

Barron

UK

61,414,470

0.03

0.26

2.03 (1.97-2.09)

Bode

USA

1122

6.2

28.8

--

De Almeida-Pititto

Meta-analysis

4.305

12.4

29.9

2.50 (1.74-3.59)

Diedisheim

France

6,314

22

26

1.81 (1.14-2.87)

Espiritu

Philippines

10,881

12.9

26.4

1.46 (1.28–1.68)

Kim

USA

10,861

--

--

1.20 (1.08-1.32)

Lazarus

Meta-analysis

14,502

--

--

1.81 (1.41-2.33)

Williamson

UK

17,278,392

0.06

0.26

1.95 (1.83-2.08)

Wu

China

44,672

2.3

7.3

--

Table 1. COVID-19 Death in T2DM: Summary of the main results.

CI, confidence interval. HR, Hazard Ratio. T2DM, type 2 diabetes mellitus

  •    *    a misprint: “S” is missed in the header of section 5: should by SARS instead of SAR

Reply: We have corrected the typo

Reviewer 2 Report

This review article discuss an interesting and important health issue “

 Diabetes mellitus and COVID-19: A narrative review”. The literature review has interesting pharmacological information, which might be utilized in clinical research. The main information of the review article is the interaction between conventional diabetic medicines and their impact on COVID-19 infection and prognosis.

However, I do suggest certain things, which need clarification to support and strengthen the article.

Points for attention are:

Title: The title indicates a focus on Diabetes mellitus and COVID-19. Literature and Data presented in this article is mostly about T2DM. Modification of the title to Type 2 diabetes mellitus (T2DM) and Covid-19 is suggested.

Abstract:

 “In addition, the most ……. discussed in this review”.  What is discussed in this review is not mentioned in the sentence that comes before it. Kindly add the missing sentence or remove “in addition….”

Introduction:

Kindly shorten the Introduction of T2DM and focus more on studies suggesting a bidirectional interaction between T2DM and COVID-19. Please mention the mechanism in brief with some supporting references.

Section 2. Epidemiology of T2DM worldwide:

  This part has not much relevance to the subject of the manuscript. Kindly merge the information in first paragraph of the Introduction section.

Section 3. T2DM as a risk factor for the prognosis of COVID-19

Kindly elaborate this part with addition of Table (data of T2DM patients compared with healthy, Covid-19, medications, complications, mortality). This is the most interesting and informative part of the article

Sub section 3.1. Epidemiologic data about DM and COVID-19 associated deaths and complications:

This paragraph is very difficult to understand. Kindly arrange the data in table form for better readability and understanding of readers.

Line81: Descrived….correct to described

Line 82: repeated “ and”. kindly reframe the sentence.

Line 84: mention full form of HR (Mentioned for the first time).

Line 99: What is EE.UU (please provide full form)?

Figure 2: Better schematic representation of possible mechanism behind T2DM and Covid 19 is required. At present there is no connection between the DM, Mechanism and Cov-19 prognosis, mortality. Please elaborate the information on mechanism.

Line 171-173: Please provide reference to this sentence.

Section 5. Newly emerging diabetic patients infected with SAR-CoV-2

Very important topic. Please provide clinical data supporting the information.

Figure 3: Lacks the interconnectivity. Please make it more informative with brief description.

Section 6. The information about common diabetes treatment needs clinical data presentation. It is descriptive with less information about the data connecting DM medications, Covid-19.

Conclusive remarks: I do think that the review contains important issues, information, interesting approaches, which can lead to proper understanding of the link between T2DM and Cov-19. The authors need to pay attention on presentation of concrete data to substantiate the interconnected effects of T2DM and Cov-19. The article needs more clinical data inputs and better representation, several improvements in the English Style, as well as formatting to improve the readability of the document, avoid spelling mistakes, Typo’s.  I consider this manuscript suitable for the publication after the suggested incorporations and improvements in the Journal of BIOMEDICINES.

Author Response

Reviewer 2
Comments and Suggestions for Authors
This review article discuss an interesting and important health issue “Diabetes mellitus and COVID-19: A narrative review”. The literature review has interesting pharmacological information, which might be utilized in clinical research. The main information of the review article is the interaction between conventional diabetic medicines and their impact on COVID-19 infection and prognosis.

However, I do suggest certain things, which need clarification to support and strengthen the article.

Points for attention are:

Title: The title indicates a focus on Diabetes mellitus and COVID-19. Literature and Data presented in this article is mostly about T2DM. Modification of the title to Type 2 diabetes mellitus (T2DM) and Covid-19 is suggested.
Reply: We have modified the title according to the reviewer’s suggestion

Abstract:

 “In addition, the most ……. discussed in this review”.  What is discussed in this review is not mentioned in the sentence that comes before it. Kindly add the missing sentence or remove “in addition….”

Reply: We have removed “In addition” from the sentence included in the Abstract.

Introduction:

Kindly shorten the Introduction of T2DM and focus more on studies suggesting a bidirectional interaction between T2DM and COVID-19. Please mention the mechanism in brief with some supporting references.
Reply: In the new version of manuscript after an initial paragraph about type 2 diabetes mellitus and the magnitude of the problem, we have included a brief review about the bidirectional link between T2DM and COVID-19 to justify our narrative review (lines 29-48): “In the last decades, type-2 diabetes mellitus (T2DM) has become a chronic metabolic disorder caused by the interaction of different genetic and environmental factors. The incidence and prevalence of patients with T2DM are increasing worldwide, even reaching epidemic values in most high- and middle-income countries (International Diabetes Federation. 10th edn). The World Health Organization (WHO) estimates that T2DM will be the seventh leading cause of death by 2030 worldwide (Figure 1) (Mathers, C.D. PLoS Med. 2006). The main reasons for this increase seem the high prevalence of obesity and the unhealthy lifestyles. Uncontrolled and prolonged T2DM can lead to serious complications, some of them being life-threatening (Baena-Díez, J.M. Diabetes Care 2016). As a result, the healthcare cost of T2DM and the related diseases is growing every year (Bommer, C. Diabetes Care 2018). Strategies to control T2DM include appropriate life-style changes as well as medication intake when necessary (Alam, S. Diabetology 2021).

T2DM could be a risk factor for developing complications in other diseases. At the onset of the pandemics, the USA Centers for Disease Control and Prevention (CDC) described that one third of patients infected with COVID-19 had comorbidities. Thus, people with at least one underlying condition account for 78% of admissions to the intensive care unit (ICU) and 94% of deaths. Diabetes mellitus was the most fre-quently reported, being the 10.9% of the cases (Chow, N. MMWR. Morb. Mortal. Wkly. Rep. 2020). In addition, a fast-growing evidence reports a bidirectional interplay between diabetes mellitus and COVID-19. Clinical data so far suggest that the severe acute respiratory syndrome coronavirus 2 (SARS-CoV-2) may result in metabolic dysregulation and in impaired glucose homeostasis (Kazakou, P. Front. Endocrinol. (Lausanne). 2022).

Section 2. Epidemiology of T2DM worldwide:

  This part has not much relevance to the subject of the manuscript. Kindly merge the information in first paragraph of the Introduction section.
Reply: In the new version of the manuscript, the contents regarding the epidemiology of diabetes mellitus have been included as part of the Introduction.

Section 3. T2DM as a risk factor for the prognosis of COVID-19

Kindly elaborate this part with addition of Table (data of T2DM patients compared with healthy, Covid-19, medications, complications, mortality). This is the most interesting and informative part of the article
Sub section 3.1. Epidemiologic data about DM and COVID-19 associated deaths and complications:
This paragraph is very difficult to understand. Kindly arrange the data in table form for better readability and understanding of readers.

Reply: We have rewritten the paragraph regarding the COVID-19 mortality risk in type 2 diabetes mellitus patients (#2 in the new version of the manuscript) to include a discussion about the role of age (lines 57-86):

“2. T2DM as a risk factor for the development and prognosis of COVID-19

Several risk factors have been associated with an increased risk of SARS-CoV2 in-fection and complication. For instance, male sex, older age, deprivation and comorbid-ities such as cardiopathy, hypertension, chronic obstructive pulmonary disease, im-munosuppression or T2DM (Baena-Díez J.M., J. Public Health (Oxf.) 2020; Baena-Díez J.M. Healthc. (Basel, Switzerland) 2021). Thus, 33.8% of 5,700 of patients with COVID-19 ad-mitted to 12 hospitals within the Northwell Health system in New York had diabetes (Richardson, S. JAMA. 2020). In addition, a random meta-analysis of 18 different studies determined that the risk of severe disease was 2.4-fold higher in patients with T2DM compared with those without (De Almeida-Pititto, B. Diabetol. Metab. Syndr. 2020); whereas another one showed a 2.6-fold higher severity risk by increased fasting blood glucose at admission (Lazarus, G. Diabetes Res. Clin. Pract. 2021).

2.1. Risk of death and complications in T2DM patients with COVID-19 

So far, different studies have shown that people with T2DM have higher risk of COVID-19 mortality compared with non-diabetic individuals (De Almeida-Pititto, B. Diabetol. Metab. Syndr. 2020; Lazarus, G. Diabetes Res. Clin. Pract. 2021; Barron, E. Lancet. Diabetes Endocrinol. 2020; Bode, B. J. Diabetes Sci. Technol. 2020; Diedisheim, M. J Clin Endocrinol Metab. 2021; Espiritu, A.I. Sci. Rep. 2021; Kim, T.S. Obesity (Silver Spring). 2021; Williamson, E.J. Nature 2020; Wu, Z. JAMA 2020). Nevertheless, Al-Salameh et al., reported that COVID-19-related death in patients with T2DM was lower than general population, but the rate of intensive care unit (ICU) admission was increased (Al-Salameh, A. Diabetes. Metab. Res. Rev. 2021). As highlighted by Diedisheim et al., the age may play a key role because after age 50 years, diabetes-related risk might be weakened by all other comorbidities or conditions associated with aging (Diedisheim, M. J Clin Endocrinol Metab. 2021) (Table 1).

In addition to the increased mortality associated with COVID-19, patients with T2DM also present with more complications from such infection, even requiring ad-mission to the intensive care unit (ICU) or dying (Richardson, S. JAMA. 2020; Barron, E. Lancet. Diabetes Endocrinol. 2020). Moreover, uncontrolled hyperglycemia can also be a risk factor for an adverse COVID-19 prognosis. An observational study including more than 1,000 patients hospitalized with COVID-19 in US showed that 40% of patients had diabetes or uncontrolled hyperglycemia at admission, and hospital mortality was four times higher for DM patients. The same study showed that mortality was seven times higher for those without pre-existing T2DM who de-veloped in-hospital hyperglycemia (Bode, B. J. Diabetes Sci. Technol. 2020). Similarly, a recent meta-analysis that included 14,502 patients confirmed these findings and showed a nonlinear relationship between fasting blood glucose at admission and severity: every mmol/L enhancement in glucose levels increased the risk of COVID-19 severity by 33% (Lazarus, G. Diabetes Res. Clin. Pract. 2021)”.

In addition, the new version of the manuscript includes a Table that summarizes the main results of the referenced studies.

Table 1. COVID-19 Death in T2DM: Summary of the main results.

Country

N

Mortality (%)

COVID-19 Death in T2DM

No T2DM

T2DM

HR (95% CI)

Al-Salameh

France

433

21.5

17.4

0.77 (0.44–1.32)

Barron

UK

61,414,470

0.03

0.26

2.03 (1.97-2.09)

Bode

USA

1122

6.2

28.8

--

De Almeida-Pititto

Meta-analysis

4.305

12.4

29.9

2.50 (1.74-3.59)

Diedisheim

France

6,314

22

26

1.81 (1.14-2.87)

Espiritu

Philippines

10,881

12.9

26.4

1.46 (1.28–1.68)

Kim

USA

10,861

--

--

1.20 (1.08-1.32)

Lazarus

Meta-analysis

14,502

--

--

1.81 (1.41-2.33)

Williamson

UK

17,278,392

0.06

0.26

1.95 (1.83-2.08)

Wu

China

44,672

2.3

7.3

--

CI, confidence interval. HR, Hazard Ratio. T2DM, type 2 diabetes mellitus

Line81: Descrived….correct to described
Reply: We have corrected the typo

Line 82: repeated “ and”. kindly reframe the sentence.
Reply: We have corrected the typo now reframed as follows: “The main risk factors identified were being male; older age and deprivation (the last two with a strong gradient)”.

Line 84: mention full form of HR (Mentioned for the first time).
Reply: We have mention the full form of Hazard Ratio (HR)

Line 99: What is EE.UU (please provide full form)?
Reply: We have corrected the abbreviated form of USA.

Figure 2: Better schematic representation of possible mechanism behind T2DM and Covid 19 is required. At present there is no connection between the DM, Mechanism and Cov-19 prognosis, mortality. Please elaborate the information on mechanism.
Reply: To clearly show the physiopathological mechanisms involved in SARS-Cov2 severity in people with T2DM, we have redesigned Figure 2.

Figure 2. Possible mechanisms behind adverse clinical outcomes of COVID-19 in people with T2DM. Adapted from Lim (Lim. Nat. Rev. Endocrinol. 2021).

Line 171-173: Please provide reference to this sentence.
Reply: We have included two references to those lines (155-157 in the new version of the manuscript):

Ref #40: Rubino, F.; Amiel, S.A.; Zimmet, P.; Alberti, G.; Bornstein, S.; Eckel, R.H.; Mingrone, G.; Boehm, B.; Cooper, M.E.; Chai, Z.; et al. New-Onset Diabetes in Covid-19. N. Engl. J. Med. 2020, 383, 789-790, doi:10.1056/NEJMC2018688.

Li, M.Y.; Ref #44:  Li, L.; Zhang, Y.; Wang, X.S. Expression of the SARS-CoV-2 cell receptor gene ACE2 in a wide variety of human tissues. Infect. Dis. poverty 2020, 9, 45. doi:10.1186/S40249-020-00662-X.

Section 5. Newly emerging diabetic patients infected with SAR-CoV-2
Reply: We have corrected the typo

Very important topic. Please provide clinical data supporting the information.

Figure 3: Lacks the interconnectivity. Please make it more informative with brief description.
Reply: To show the interconnectivity in the physiopathological pathway, we have included a new Figure 3 in the new version of the manuscript adapted from Apicella M. Lancet Diabetes Endocrinol. 2020.

Figure 3. Potential pathogenic mechanisms of SARS-CoV-2 infection underlying metabolic deregulation and homeostasis alteration. Adapted from Apicella et al.

Section 6. The information about common diabetes treatment needs clinical data presentation. It is descriptive with less information about the data connecting DM medications, Covid-19.
Reply: We have included clinical data, when available, for the drug listed in the manuscript.

5.1. Metformin (lines 212-215):

In addition, a meta-analysis estimated a risk reduction associated with this treatment in individuals with T2DM and SARS-CoV2 infection [Odds ratio = -0.37; 95% confidence interval (-0.59; -0.16)] (Kan, C. Front. Endocrinol. (Lausanne). 2021).

5.2. Insulin (lines 233-236)

In fact, insulin treatment was associated with increased systemic inflammation and increased damage to vital organs, suggesting that insulin therapy for patients with COVID-19 and T2DM needs to be used with caution (27.2% versus 3.5%; adjusted Hazard Ratio = 5.38; 95% CI [2.75–10.54]) (Yu, B. Cell Metab. 2021).

5.3. SGLT2 (Sodium-glucose cotransporter-2 inhibitors) (lines 248-250)

The meta-analysis performed by Nguyen et al., showed a significant mortality reduction in individuals with T2DM treated with SGLT2 inhibitors previous to hospital admission (OR = 0.60 [0.40–0.88]) (Nguyen, N.N. Metabolism. 2022).

5.4. Sulfonylureas (lines 260-262)

On the one side, a systematic review and meta-analysis highlight the potential for sul-fonylurea treatment to reduce the risk of mortality in T2DM patients with COVID-19 (OR = 0.80; 95% CI [0.66; 0.96]) (Kan, C. Front. Endocrinol. (Lausanne). 2021).

5.6. Dipeptidyl peptidase-4 (DPP4) inhibitors (lines 285-291)

A metanalysis including 10 observational studies found that DDP-4 inhibitors reduce the risk for COVID-19-related mortality by 50% (Patoulias, D. Endocrinol. Metab. (Seoul, Korea) 2021). Other metanalysis also support the hypothesis that DDP-4 inhibitors could have a protective effect on COVID-19 (OR = 0.58; 95% CI [0.34-0.99]) (Yang, Y. PLoS One 2021). Finally, the reduction in mortality was marginally sig-nificant in the metanalysis performed by Kan et al. (OR = 0.72; 95% CI [0.51-1.01]) (Kan, C. Front. Endocrinol. (Lausanne). 2021). Thus, further research is necessary to evaluate the role of DPP4 inhibitors in patients with T2DM and COVID-19.

5.7. Glucagon-like peptide 1 (GLP-1) receptor agonist (lines 301-303)

A recent Bayesian Network Metanalysis GLP-1 receptor agonist treatment was liked to a decrease in COVID-19-related mortality in T2DM individuals compared to non-users (OR = 0.91; 95% CI [0.84; 0.98]) (Chen, Y. Front. Endocrinol. (Lausanne). 2022).

Conclusive remarks: I do think that the review contains important issues, information, interesting approaches, which can lead to proper understanding of the link between T2DM and Cov-19. The authors need to pay attention on presentation of concrete data to substantiate the interconnected effects of T2DM and Cov-19. The article needs more clinical data inputs and better representation, several improvements in the English Style, as well as formatting to improve the readability of the document, avoid spelling mistakes, Typo’s.  I consider this manuscript suitable for the publication after the suggested incorporations and improvements in the Journal of BIOMEDICINES.

Reply: Thank you for the comments. We have made a huge effort to improve the quality of the manuscript (clinical data inputs and representation). Additionally, we have reviewed the English style in depth.
